# KSK-74: Dual Histamine H_3_ and Sigma-2 Receptor Ligand with Anti-Obesity Potential

**DOI:** 10.3390/ijms23137011

**Published:** 2022-06-24

**Authors:** Kamil Mika, Małgorzata Szafarz, Monika Zadrożna, Barbara Nowak, Marek Bednarski, Katarzyna Szczepańska, Krzysztof Pociecha, Monika Kubacka, Noemi Nicosia, Izabela Juda, Katarzyna Kieć-Kononowicz, Magdalena Kotańska

**Affiliations:** 1Department of Pharmacological Screening, Jagiellonian University Medical College, Medyczna 9, 30-688 Cracow, Poland; kamil.mika@doctoral.uj.edu.pl (K.M.); marek.bednarski@uj.edu.pl (M.B.); monika.kubacka@uj.edu.pl (M.K.); noemi.nicosia92@gmail.com (N.N.); 2Department of Pharmacokinetics and Physical Pharmacy, Jagiellonian University Medical College, Medyczna 9, 30-688 Cracow, Poland; malgorzata.szafarz@uj.edu.pl (M.S.); k.pociecha@uj.edu.pl (K.P.); 3Department of Cytobiology, Jagiellonian University Medical College, Medyczna 9, 30-688 Cracow, Poland; monika.zadrozna@uj.edu.pl (M.Z.); barbara.anna.nowak@uj.edu.pl (B.N.); izabela.juda@student.uj.edu.pl (I.J.); 4Technology and Biotechnology of Medical Remedies, Faculty of Pharmacy, Jagiellonian University Medical College, Medyczna 9, 30-688 Cracow, Poland; szczepanskatarzyna@gmail.com (K.S.); katarzyna.kiec-kononowicz@uj.edu.pl (K.K.-K.); 5Department of Medicinal Chemistry, Maj Institute of Pharmacology Polish Academy of Sciences, Smętna 12, 31-343 Cracow, Poland; 6Foundation “Prof. Antonio Imbesi”, University of Messina, Piazza Pugliatti 1, 98122 Messina, Italy; 7Department of Chemical, Biological, Pharmaceutical and Environmental Sciences, University of Messina, Viale Palatucci, 98168 Messina, Italy

**Keywords:** histamine H_3_ receptor ligands, sigma-2 receptor ligands, palatable diet, excessive eating model, obesity

## Abstract

Many studies involving compounds that enhance histamine release, such as histamine H_3_ receptor (H_3_R) antagonists, have shown efficacy in inhibiting weight gain, but none have passed clinical trials. As part of the search for H_3_ receptor ligands that have additional properties, the aim of this study is to evaluate the activity in the reduction in weight gain in a rat model of excessive eating, as well as the impact on selected metabolic parameters, and the number and size of adipocytes of two new H_3_R antagonists, KSK-60 and KSK-74, which also exert a significant affinity at the sigma-2 receptor. Compounds KSK-60 and KSK-74 are homologues and the elongation of the distal part of the molecule resulted in an approximate two-fold reduction in affinity at H_3_R, but simultaneously an almost two-fold increase in affinity at the sigma-2 receptor. Animals fed palatable feed and receiving KSK-60 or KSK-74 both at 10 mg/kg b.w. gained significantly less weight than animals in the control obese group. Moreover, KSK-74 significantly compensated for metabolic disturbances that accompany obesity, such as an increase in plasma triglyceride, resistin, and leptin levels; improved glucose tolerance; and protected experimental animals against adipocyte hypertrophy. Furthermore, KSK-74 inhibited the development of inflammation in obesity-exposed adipose tissue. The in vivo pharmacological activity of the tested ligands appears to correlate with the affinity at the sigma-2 receptors; however, the explanation of this phenomenon requires further and extended research.

## 1. Introduction

The World Health Organization (WHO) considers obesity to be one of the most serious public health concerns of the present century. The WHO has estimated that, globally, there are more than 650 million people with obesity (BMI ≥ 30 kg/m^2^) and, in 2020, over 39 million children under the age of 5 years were overweight or obese [1]. Further concerns arise from the strong correlation between obesity or being overweight and the progression of cardiovascular and gastrointestinal diseases and diabetes, along with the increased risk of developing pancreatic, kidney, colorectal, and gallbladder cancer, as well as musculoskeletal disorders and infections [2,3]. Considering its high impact not only on health, but also on the economy and society, over the past few years, the pharmaceutical industry and many research groups have focused on the development of an effective and safe drug to fight this global epidemic. The numerous pharmaceutical anti-obesity treatment options (e.g., dinitrophenol, fenfluramine, fenfluramine-phentermine, sibutramine, and rimonabant) were associated with severe side effects and had to be removed from the market by both the Food and Drug Administration (FDA) and by the European Medicine Agency (EMA). In the treatment of obesity, both of these agencies (EMA and FDA) have only approved the use of Orlistat, Liraglutide, and the combination of Bupropion and Naltrexone [3].

Considering the involvement of the histaminergic system in food intake and body weight regulation through the interaction with histamine receptors in the CNS, many recent studies have focused on the histamine H_3_ receptor (H_3_R) as a potential target for anti-obesity therapies [4,5,6]. H_3_R is a presynaptic autoreceptor abundantly expressed in the CNS and modestly found peripherally. Its activation leads to a negative feedback modulation of histamine synthesis and inhibition of its release from histaminergic neurons. Furthermore, histamine postsynaptic H_3_R has been shown to modulate the release of other neurotransmitters, including dopamine, acetylcholine, serotonin, glutamate, substance P, norepinephrine, and γ-aminobutyric acid [4]. To date, many studies involving compounds that enhance histamine release, such as H_3_R antagonists, have shown efficacy in inhibiting weight gain [5,6,7,8,9]. Histamine also increases the lipolysis of white adipose tissue, thus altering peripheral metabolism [10]. This fact may justify the strong interest in the search for new, effective, and safe therapeutic agents to treat obesity among H_3_R ligands.

To date, numerous molecules with antagonist/inverse agonist properties have been synthesized from the proposed H_3_R pharmacophore structure that contains a basic moiety—mostly a tertiary amine substituted by a linking alkyl group, often incorporating another functionality (frequently polar) [5,11,12]. Early preclinical studies revealed the efficacy of many H_3_ receptor antagonists in reducing body weight in rodent models; however, only SCH-497079 and HPP404 were selected to enter phase II clinical trials. Among these, the antagonist SCH-497070 has recently completed the trials. Unfortunately, both molecules were not considered for further development due to the side effects [13]. Furthermore, A-331440, a selective H_3_R antagonist, was found to be useful in decreasing body weight. The administration of the highest dose of A-331440 in mice fed a high-fat diet induced weight loss comparable to mice fed a low-fat diet along with lower leptin levels and normalized insulin tolerance. Despite its effectiveness, it was excluded from clinical trials due to genotoxicity [14].

Promising results were obtained from studies on Betahistine, a H_3_R antagonist and H_1_R agonist (registered for the treatment of vertigo and Meniere’s disease). Betahistine induced significant weight loss in an obese animal after olanzapine treatment [15,16]. In addition, in our previous work, we described the effect of the repeated administration of Betahistine on body weight in a rat model of excessive eating. We showed that long-term Betahistine administration slows weight gain and increases high-density lipoprotein (HDL) levels [17]. A clinical study compared the effect of Betahistine with Orlistat^®^ in obese adults [18]. The experiment did not show significant differences between the two molecules in reducing body weight and waist circumference. Apart from Betahistine, Pitolisant is the second approved (in narcolepsy) histamine H_3_R antagonist. Kotańska et al. (2018) investigated the effect of the antagonist/inverse agonist, Pitolisant (registered under the name of Wakix^®^), on body weight and metabolic disturbances in a model of induced obesity in mice. Pitolisant administered intraperitoneally (i.p.) at a dose of 10 mg/kg b.w. for 14 days showed a positive influence on body weight, glucose tolerance, and lipid profile [19].

In our previous publication, we found that KSK19, one of the most active and selective ligands of H_3_ receptors, exerted a favorable impact on body weight, after multiple administrations at a dose of 15 mg/kg b.w., in the mice obesity model [20]. Considering its promising anti-obesity properties, our research group chose the KSK19 lead structure as a reference to develop new selective H_3_R ligands. Our recent study demonstrated high efficacy in inhibiting weight gain in a model of excessive eating and favorable pharmacokinetic properties for four of the ligands tested: KSK-61 and KSK-63, KSK-59 and KSK-73 [21,22]. Based on our recent work, we included two new histamine H_3_R antagonists in our study, KSK-60 and KSK-74 (Figure 1) [12], which are structural analogs of compounds KSK-59 and KSK-73. We used phentermine as the reference compound, which is currently registered in some countries (also in combination with Topiramate) for the treatment of obesity [23]. Our compounds have also been shown to exert a significant affinity at the sigma-2 receptor [24]. Interestingly, looking at the results for the compounds KSK-60 and KSK-74, which are the subject of this study, the elongation of the distal part of the molecules (acetyl vs. propionyl derivatives, Figure 1) resulted in an approximate two-fold reduction in affinity at H_3_R, but simultaneously an almost two-fold increase in affinity at the sigma-2 receptor. Therefore, a comparison of the activity in the reduction in weight gain in the model of excessive eating and the impact on selected metabolic parameters, as well as the number and size of adipocytes after the administration of these two compounds, can provide valuable information on the importance of this affinity.

## 2. Results

### 2.1. The Intrinsic Activity at H_3_R

The intrinsic activity towards H_3_R of all compounds tested was examined using two commercial methods, and the obtained IC_50_ values, despite the slight differences, were comparable. The compound KSK-60 proved to be a more potent antagonist of H_3_R, with an IC_50_ value ranging from 0.8 to 1.5 nM, depending on the assay method. The compound KSK-74 was slightly less active in blocking H_3_R, and the IC_50_ value for this compound ranged from 23 to 32 nM depending on the assay method (Table 1, Figure 2). Summing up, both tested compounds showed significant antagonist properties for H_3_R, with KSK-60 acting more effectively.

### 2.2. Effect of KSK-60, KSK-74, or Phentermine on Body Weight and Caloric Intake

A higher weight gain was observed in rats in the palatable diet + vehicle group than in rats in the Standard diet + vehicle group (*p* < 0.01, *p* < 0.001, Figure 3a,b). Animals fed palatable feed and receiving KSK-60 or KSK-74 both at 10 mg/kg b.w. gained significantly less weight than animals in the palatable diet + vehicle group (*p* < 0.05, *p* < 0.01, respectively, Figure 3b). No differences in body weight were observed in the palatable diet + KSK-60 or palatable diet + KSK-74 groups (both 1 mg/kg) compared to the palatable diet + vehicle group. Rats fed palatable feed and treated with phentermine gained less weight than rats receiving palatable diet + vehicle (*p* < 0.01, Figure 3b). In rats fed a palatable diet and treated with compounds KSK-60 or KSK-74 (10 mg/kg), we recorded a significant inhibition of body weight gain around the 10th day of the experiment, and this condition persisted until the 28th day (the end) of the experiment. The results are shown in Figure 3c,d.

KSK-74 or KSK-60 administered i.p. for 28 days at a dose of 10 mg/kg b.w. or phentermine administered at a dose of 7 mg/kg b.w. did not significantly influence the amount of calories consumed by animals in the test groups compared to the control group fed palatable feed (Figure 4). In KSK-74- or phentermine-treated groups, rats ate slightly less calories (lack of significance vs. both control groups). Rats treated with KSK-74 at a dose of 10 mg/kg b.w. consumed significantly less milk in the first and second weeks and significantly less peanuts in the fourth week of treatment vs. control group fed palatable feed (Figure 5).

### 2.3. Effect of KSK-60, KSK-74, or Phentermine on Fat Pads and Plasma Triglyceride Levels

A statistically higher amount of fat pads in the peritoneal cavity was observed in the palatable diet + vehicle group than in the standard diet + vehicle group (*p* < 0.05, Figure 6a). In the palatable diet + KSK-74 (10 mg/kg) group, fewer fat pads were observed than in the palatable diet + vehicle group (*p* < 0.001, Figure 6a). Animals fed palatable feed and receiving KSK-60 at 10 mg/kg or phentermine at 7 mg/kg had fewer peritoneal fat pads compared to animals receiving vehicle and fed palatable feed, since there was no statistically significant difference between the amount of adipose tissue in these groups vs. the control group fed standard feed (Figure 6a). Compared to the standard diet + vehicle group, higher plasma triglyceride levels were observed in rats fed palatable feed and receiving vehicle (*p* < 0.01, Figure 6b). Animals fed palatable feed and receiving KSK-74 at 10 mg/kg or phentermine had lower plasma triglyceride levels than rats from the palatable diet + vehicle group (*p* < 0.05, *p* < 0.01, respectively, Figure 6b). No differences in plasma triglyceride levels were observed between the palatable diet + vehicle group and the palatable diet + KSK-60 group (10 mg/kg) (Figure 6b).

### 2.4. Effect of KSK-60, KSK-74, or Phentermine on the Numerical Density of Adipocytes

Morphometric analysis of adipose tissue showed a decrease in the number of adipocytes per 0.1 mm^2^ of cross-sectional area both in the palatable diet + vehicle and the palatable diet + KSK-60 (10 mg/kg) groups compared to the standard diet + vehicle group (17.69 ± 1.28 vs. 28.73 ± 1.54, *p* < 0.05 and 18.02 ± 1.38 vs. 28.73 ± 1.54, *p* < 0.05, respectively). The highest number of adipocytes per unit area was observed in the adipose tissue of animals in the palatable diet + KSK-74 (10 mg/kg) group (34.78 ± 4.78) and was almost two times higher than in the palatable diet + vehicle (*p* < 0.05) and the palatable diet + KSK-60 (10 mg/kg) groups (*p* < 0.05), confirming the protective effect of the KSK-74 compound against adipocyte hypertrophy. The mean number of adipocytes in the adipose tissue of the palatable diet + phentermine (7 mg/kg) group was more similar to the results of the palatable diet + vehicle group than to the results of the non-obese control group, but these differences were not significant in both cases (Figure 7).

### 2.5. Effect of KSK-60, KSK-74, or Phentermine on the Presence of Pathological Features of Adipose Tissue Inflammation

Histological analysis was performed to identify the pathological features of adipose tissue inflammation. Many inflammatory cells infiltrated the area of adipocytes and the perivascular area, mainly in the adipose tissue of rats from the palatable diet + vehicle and palatable diet + KSK-60 (10 mg/kg) groups. On the other hand, slightly less inflammatory infiltrates, rather in the form of local leukocyte clusters, were observed in tissues of animals from the palatable diet + phentermine (7 mg/kg) and the palatable diet + KSK-74 (10 mg/kg) groups. Masson’s trichrome staining of adipose tissue sections revealed a frequent occurrence of trichrome-positive fibrotic streaks in tissue from animals of all groups, except rats from the standard diet + vehicle group. In addition, no leukocyte infiltrates were observed in this group (Figure 8).

In the adipose tissue of rats in the palatable diet + phentermine (7 mg/kg) group, our attention was focused on the marked capillary congestion and frequent appearance of crown-like structures formed by mononuclear cells surrounding a presumably dead adipocyte (Figure 9).

All these observations were further confirmed by the histological scoring of inflammation and presented as the inflammatory index. The number of rats in each group, with the appropriate histological grade for the independently assessed components of the inflammatory index is presented in Table 2. The inflammatory index of the adipose tissue of the studied animals was statistically significantly higher in the palatable diet + vehicle (4.33 ± 0.42, *p* < 0.05), palatable diet + KSK-60 (10 mg/kg) (4.5 ± 0.80, *p* < 0.05), and palatable diet + phentermine (7 mg/kg) (4.16 ± 0.48, *p* < 0.05) groups compared to the standard diet + vehicle group (0.5 ± 0.34) (Figure 8f).

### 2.6. Effects of KSK-60, KSK-74, or Phentermine on IL-6 and MCP-1 Levels in Adipose Tissue

There were no significant changes in IL-6 levels in adipose tissue between the standard diet + vehicle and the palatable diet + vehicle control groups. IL-6 levels were reduced in adipose tissue only in animals fed palatable feed and receiving KSK-74 at 10 mg/kg, compared to the group receiving a palatable diet and vehicle (*p* < 0.01, Figure 10a). Higher levels of MCP-1 were observed in the adipose tissue of rats in the palatable diet + vehicle group compared to the adipose tissue of rats in the standard diet + vehicle group (*p* < 0.001, Figure 10b). The administration of KSK-74 at a dose of 10 mg/kg b.w. decreased the level of MCP-1 in the rat’s adipose tissue, compared to the rats that received palatable feed and vehicle (*p* < 0.001, Figure 10b).

### 2.7. Effect of KSK-60, KSK-74, or Phentermine on Leptin and Resistin Levels in Adipose Tissue

There were significant changes in leptin and resistin levels in adipose tissue between rats in control groups that received different diets (standard or palatable) and vehicle (*p* < 0.05, Figure 11a,b). Compared to the palatable diet + vehicle group, reduced levels of leptin and resistin were observed in adipose tissue from animals that received i.p. KSK-74 10 mg/kg (*p* < 0.001, *p* < 0.01, Figure 11a,b). Phentermine did not influence the concentration of leptin or resistin.

### 2.8. Glucose Tolerance Test

Blood glucose levels at 30 and 60 min after glucose load in rats receiving palatable feed and vehicle were significantly higher, compared to glucose levels determined at the same time points in rats receiving standard feed and vehicle. In the groups treated with KSK-60 or KSK-74 at a dose of 10 mg/kg, glucose levels were significantly lower at 30 or/and 60 min after glucose load than in control animals fed palatable feed (Figure 12). As shown in Figure 12d, AUC decreased significantly after treatment with KSK-60 or KSK-74, compared to the control value observed in rats fed high-calorie feed.

### 2.9. Effect of KSK-60, KSK-74, or Phentermine on Spontaneous Activity

Spontaneous activity was measured after the 1st (single dose) and 27th (multiple doses) administrations of the tested compounds. The compound KSK-60 after a single administration at a dose of 10 mg/kg b.w. reduced spontaneous activity at the 17th hour of measurement, while, after the repeated administration of this compound, we observed a reduction in spontaneous activity at the 15th hour of measurement. A single injection of phentermine decreased spontaneous activity at the 17th hour of measurement, while, after multiple administrations of this compound, we recorded an increase in spontaneous activity at the 2nd hour of measurement. The single and repeated administrations of compound KSK-74 did not affect spontaneous activity in rats (Figure 13).

### 2.10. Effect of KSK-60, KSK-74, or Phentermine Administration on Plasma Corticosterone

There was no significant change in plasma corticosterone levels between different groups of rats. In this experiment, the tested compounds had no effect on the plasma corticosterone levels of rats. Its plasma concentrations ranged between 120–155 pg/mL.

### 2.11. Pharmacokinetic Analysis

The pharmacokinetic (PK) parameters calculated using non-compartmental analysis based on the concentration-time data after the single i.p. administration of the investigated compounds at a dose of 10 mg/kg b.w. are presented in Table 3. The maximal plasma concentration was attained by both compounds at the first sampling point (5 min after administration), and the values were almost identical (109 vs. 105 µg/L for KSK-60 and KSK-74, respectively). Both compounds also had identical half-lives equal to 2.7 h. However, the compound KSK-74 had much higher clearance due to the larger volume of distribution (272 vs. 513 L/kg for KSK-60 and KSK-74, respectively), indicating that it much easier, compared to the KSK-60, crossed the biological membranes, and distributed throughout the body.

## 3. Discussion

In this study, we performed preliminary pharmacological experiments to determine the potential anti-obesity properties of two H_3_R ligands, KSK-60 and KSK-74, which are also ligands of sigma receptors. The effect of their chronic administration on body weight, selected metabolic parameters, and spontaneous activity of rats was investigated in the model of excessive eating of preferential feed. We also conducted studies that evaluated the intrinsic activity and selected pharmacokinetic parameters of these compounds. To take a closer look at the action of our selected ligands, we performed histopathological studies showing the numerical density of adipocytes and the pathological features of inflammation in peritoneal adipose tissue.

In our previous work, we published the results for their structural analogues, compounds KSK-59 and KSK-73, which differ only in the distal part of the molecules (acetyl versus propionyl derivatives, Figure 1) [22]. It should be noted that compound KSK-73 prevented weight gain and metabolic disturbances more than compound KSK-59, which is partially consistent with our current results, in which KSK-74 is also more effective than KSK-60. Although both compounds similarly inhibited weight gain in the excessive eating model, for the most part, only compound KSK-74 significantly compensated for metabolic disturbances that accompany obesity.

Comparing the structures of tested compounds, the eight carbon-chain homologues were characterized by higher efficacy in vivo for both acetyl and propionyl derivatives. However, our previous work showed that compounds with the highest affinity for H_3_R were not the most active in vivo [20,21]. Therefore, we suspect that the influence on body weight gain and metabolic activity of these ligands may be additionally related to another non-histaminic mechanism of action. Furthermore, the pharmacokinetic properties of compounds KSK-73 and KSK-74 are more favorable compared to their homologues KSK-59 and KSK-60. In particular, they have a much larger volume of distribution, which could indicate their better ability to freely cross biological membranes and distribute throughout the body [22]. Additionally, according to Lipinski’s “Rule of Five” (molecular weight ≤ 500; LogP ≤ 5; hydrogen-bond donors ≤ 5; hydrogen-bond acceptors ≤ 10; number of rotatable bonds ≤ 10), compounds KSK-60 and KSK-74 are likely to have good absorption and permeability. As for the good CNS penetration, compound KSK-60 meets 4 (molecular weight ≤ 400; LogP ≤ 5; hydrogen-bond donors ≤ 3; hydrogen-bond acceptors ≤ 7—calculated by CompuDrug Pallas System, USA) out of 4 criteria, while KSK-74 meets 3 of them (molecular weight is > 400) [25]. Based on the theoretical assumptions, it is likely that both of them are able to cross the blood–brain barrier. However, further studies are necessary to verify this hypothesis. Interestingly, the in vivo pharmacological activity of the tested ligands appears to correlate with the affinity at sigma-2 receptors, and the explanation of this phenomenon requires further extended research. Both the dual affinity at histamine H_3_/sigma-2 receptors and the significant in vivo activity of the KSK-73 and KSK-74 ligands give hope for the discovery of unique compounds among a wide variety of histamine H_3_R antagonists. Therefore, compounds KSK-73 and KSK-74 have been chosen as new lead structures for the development of H_3_/sigma-2 receptors ligands (the affinity ratios are 0.68 and 0.59 for KSK-73 and KSK-74, respectively) with anti-obese activity, and the mechanism of their action would be a subject of further studies.

The excessive eating model used in our study allows us to determine whether the studied compounds affect body weight in animals that have free access to high-calorie foods. It perfectly illustrates the unnecessary high caloric intake by overeating freely available tasty products rich in sugar and fat. It shows that the unlimited availability of tasty foods prompts the body to consume that particular food, even when extra calories are not needed and when such behavior can lead to a significant increase in body weight and the development of metabolic disorders over a short period of time. Animals have access to certain high-calorie foods, such as peanuts, cheese, milk with increased fat content, and chocolate, but also to the standard feed. Most importantly, feeding is not in any way forced [26,27]. Special diet not only induces obesity, but also contributes to metabolic disorders specifically caused by this condition. Therefore, such a model of excessive eating is considered to be an apparent imitation of human models of the obesogenic diet [28].

The effect of H_3_ receptor antagonists on food intake has already been described in the literature [20,21,22,29]. Since treated rats consumed a similar amount of calories to the control rats fed palatable feed, we speculated that, other than a reduction in caloric intake, mechanisms resulting in the lower weight gain in these animals are involved. We obtained similar results earlier when testing other ligands from this group of compounds, namely, not all of them decreased the amount of calories consumed [21,22]. We reported that i.p. administration of H_3_R antagonists KSK-60 or KSK-74 effectively and dose-dependently inhibited weight gain, and that this inhibition occurred parallel to the inhibition of fat gain. Interestingly, the animals treated with the tested compounds gained less weight than the animals that received phentermine, a drug used to treat obesity. We also observed that the most active ligands we described to date in the model of excessive eating were characterized by a significant affinity at the sigma-2 receptor [20,21,22]. We can speculate that the inhibition of weight gain may be related to H_3_R and sigma-2 receptors, making them dual ligands. However, more studies are needed to clarify this issue.

The obesity problem is primarily related to an increase in the volume of peritoneal fat. Previous studies have shown that histamine affects body weight not only by inhibiting appetite, but it also regulates visceral fat volume [4]. This fact is probably related to neuronal histamine activity, which can accelerate adipose tissue lipolysis by activating the sympathetic nervous system. The lipolysis process leads to the release of free fatty acids and glycerol, which can be used as energy substrates for the body and thus contribute to the maintenance of adequate energy homeostasis [30]. One of the main features of obesity is the abnormal metabolism of circulating lipids [31]. The excess calories supplied with food cause unused triglycerides to be stored in adipose tissue. In addition, in our experiment, we observed a significantly lower intraperitoneal fat gain in animals treated with the tested compounds, especially KSK-74, which was closely correlated with the lower body weight gain. Moreover, we noticed that the reduction in plasma triglyceride levels was also correlated with lower fat mass. During the development of obesity, adipose tissue can grow by hypertrophy, which is an increase in the size of adipocytes, or by hyperplasia, which is an increase in the number of adipocytes due to the recruitment of new cells [32]. The volume of adipocytes reflects the balance between lipogenesis and lipolysis, while the number of adipocytes reflects the balance between the proliferation, differentiation, and apoptosis of preadipocytes and the adipocytes [33]. In the early stages of obesity development, when excess calories are supplied to the body, there is an overgrowth of adipocytes that secrete adipokines. In a subsequent step, adipokines stimulate the formation of additional preadipocytes, which differentiate into mature adipocytes that protect the organism against some of the adverse metabolic consequences of obesity [34]. This concept was supported by a Wang et al. in a study conducted on mice fed a high-fat diet. These animals showed signs of visceral hypertrophy within 1 month [35]. We observed a similar phenomenon in our experiment. In animals in the control group fed palatable feed, we recorded significant adipocyte hypertrophy. Interestingly, in the group of animals that received KSK-74, we observed an increase in their numerical density, but not in size, which resembled the condition and shape of adipocytes from adipose tissue from rats fed a standard diet. Therefore, we can speculate that KSK-74 protects against the development of hypertrophy. On the other hand, phentermine did not show any positive effect on the numerical density of adipocytes. Given the current reports that there is a strong association between small adipocytes and increased insulin sensitivity, we can also speculate that the tested ligands for H_3_/sigma-2 receptors promote the formation of new adipocytes, protecting against the adverse consequences of obesity, including insulin resistance [36].

Many factors contribute to the formation of obesity, one of which is the hormonal state of the body. Information signals from the body’s periphery can be divided into those consisting of short-term signals that are generated during a meal and originate mainly from the gastrointestinal tract, and those that involve long-term signals provided by hormones that determine the body’s energy stores [37]. Among the hormones produced by adipose tissue, leptin and resistin deserve special attention. Their levels are closely correlated with the amount of adipose tissue [38,39].

Leptin is otherwise known as the satiety hormone, due to its main functions of regulating food intake and energy balance in the body. An increase in leptin levels sends a signal to the hypothalamus that energy reserves are full, and consumption can be completed; the effect is a reduction in appetite. Furthermore, high levels of leptin also increase energy expenditure (acceleration of metabolism). People who are overweight or obese have higher levels of leptin in plasma and adipose tissue [40]. In obesity, when leptin levels are high, the number of receptors for leptin is gradually reduced and their sensitivity to this hormone decreases. In this situation, the satiety signal that should normally inhibit appetite does not reach the brain [41,42]. In our experiment, in animals fed high-calorie products, we induced a state of elevated leptin levels. Animals treated with KSK-60 or KSK-74 maintained leptin levels at the same level as rats in the control group fed standard feed. Only in animals receiving KSK-74 did we observe lower leptin levels compared to rats in the control group fed palatable feed. This fact allowed us to conclude that the compounds we studied maintained leptin at an adequate level that allowed for the proper functioning of the hunger and satiety centers. These findings also allowed us to confirm that leptin levels were in direct correlation with levels of adipose tissue. Thus, maintaining optimal body fat levels is the best way to ensure physiological leptin levels and control excessive appetite and the development of obesity.

As with leptin, elevated levels of resistin are also observed in the state of excess adipose tissue. Ongoing mild inflammation is observed in overweight or obese individuals. Previous studies have shown that resistin levels correlate significantly with inflammatory markers. It is probably related to the fact that some proinflammatory cytokines, e.g., IL-6 or TNF-α, stimulate the expression of the resistin gene [43,44]. In our study, we noticed a similar relationship. In addition, elevated resistin levels can not only be a biomarker, but also a pathogenic factor for inflammatory diseases.

As mentioned above, a diet rich in high-calorie fatty foods may be the cause of mild chronic inflammation [45,46]. This condition can lead to changes in peripheral signaling associated with the insulin receptor, thereby reducing the sensitivity to insulin-mediated glucose release. These events result in elevated insulin and fasting glucose levels and decreased glucose tolerance, which may indicate that insulin resistance is developing [47]. In the model that we used, we also observed impaired glucose tolerance in the control group that had access to the palatable diet. However, a significant improvement in glucose tolerance was observed in experimental animals treated with the tested compounds. We suspected that this may be due to the fact that the experimental animals did not develop full obesity or to the effect of the tested ligands on H_3_R present in pancreatic β cells responsible for insulin secretion and blood glucose regulation [48].

In chronic inflammation, also known as metabolic inflammation, the immune cells responsible for infiltration of proliferating adipose tissue gradually turn into resident adipose tissue macrophages (ATMs). ATMs are classified by the expression of different markers, including pro-inflammatory ones, such as IL-6, MCP-1 TNF-α, CD11c, and iNOS, among others [49]. IL-6 is a pro-inflammatory cytokine responsible for the regulation of inflammation and the defense mechanisms of the body. The level of IL-6 increases significantly in adipose cells in obesity, which is due to the presence of inflammation [50]. In our study, there were no significant differences in IL-6 level in adipose tissue between the control group fed standard and palatable feeds, but there was some upward trend in the level of this mediator in the group fed palatable feed. Perhaps a longer experiment would result in a higher increase in this pro-inflammatory interleukin. Only in animals treated with compound KSK-74 did we observed a decrease in the IL-6 level in adipose tissue. These observations correlate with the results on the severity of inflammation in the tissues tested. One of the first CC chemokines discovered and best described is monocyte chemoattractant protein-1 (MCP-1). Obese people have been shown to have higher levels of this chemokine in adipose tissue than non-obese people [51,52]. In our model, we observed a similar phenomenon of elevated levels of this chemokine in rats from a control group fed palatable feed. Similar to IL-6 levels, compound KSK-74 decreased MCP-1 levels in adipose tissue and therefore nullified the formation of inflammation. To date, many studies indicate that obesity caused by a high-calorie diet leads to the induction of multiple inflammatory pathways [53]. The inflammation that develops is a consequence of the hypertrophy of adipose tissue cells and the damage to cellular structures [54]. Consistent with the reports in the literature mentioned above, we also observed ongoing inflammation among control animals fed palatable food. Of the compounds tested, only compound KSK-74 inhibited the development of inflammation in obesity-exposed adipose tissue and the value of the inflammation index was comparable to the standard control. These results support the current reports that adequate adipogenesis and hyperplasia, or the ability to distribute fat between newly formed adipocytes without the need for significant adipocyte hypertrophy, attenuates inflammation and subsequent insulin resistance [55]. When using phentermine, a drug currently registered for the treatment of obesity, we did not observe significant differences.

Severe, especially chronic, stress can cause significant weight loss. The defense response to stress is the increased secretion of hormones, such as cortisol, in humans. In previous animal studies, the introduction of a stress factor has been shown to cause changes in locomotor activity and increased corticosterone secretion [56,57]. To confirm that our results of decreased weight gain are not related to stress or changes in spontaneous activity, we determined plasma corticosterone levels in experimental animals and monitored spontaneous activity after initial and chronic administration of the tested ligands. In our study, we did not observe changes in corticosterone levels in both control groups and between animals that received the test compounds. We also did not show any significant effect on spontaneous activity, allowing us to conclude that weight loss was not caused by the stress factor.

LIMITATIONS. The main limitation of this research is that it did not directly indicate that the described actions were really related to the influence on the histamine H_3_ or sigma receptors. Although we know from the earlier cited studies and from the intrinsic activity studies presented in this manuscript that the tested compounds are ligands for these receptors, further research is needed to confirm the mechanism of their activity. The second limitation was that the number of animals used in this study was small, and the effect on calorie intake requires further research.

## 4. Materials and Methods

### 4.1. Drugs, Chemical Reagents, and Other Materials

The tested compounds (KSK-60 and KSK-74) were synthesized at the Department of Technology and Biotechnology of Drugs, Faculty of Pharmacy, Jagiellonian University Medical College, Cracow, Poland. The identity and purity of the final product were assessed by NMR and LC-MS techniques (the minimum purity was greater than 95%). For both pharmacokinetic and pharmacological studies, KSK-60 and KSK-74 (1 or 10 mg/kg) were suspended in 1% Tween 80 and the volume was adjusted to 1 mL/kg. Heparin was delivered from Polfa Warszawa S.A. (Warsaw, Poland), thiopental sodium was obtained from Sandoz GmbH, (Kundl, Austria), and phentermine was obtained from the National Measurement Institute (Pymble, Australia); (R)-alpha-methylhistamine, thioperamide, and clobenpropit were obtained from Merck KGaA (Darmstadt, Germany).

### 4.2. In Vitro Studies—Intrinsic Activity at the Histamine H_3_ Receptor

Intrinsic activity studies were performed using two methods, Aequoscreen and LANCE Ultra cAMP assays, according to the manufacturer of the ready-to-use cells with stable expressions of the H_3_ histamine receptor (Perkin Elmer). In both assays (R)-alpha-methylhistamine dihydrobromide was used as the reference agonist. thioperamide or clobenpropit were used as the reference antagonists in LANCE Ultra cAMP and Aequoscreen assays, respectively.

The Aequoscreen technology uses recombinant cell lines with stable co-expression of apoaequorin and a GPCR as a system to detect activation of the receptor, following the addition of an agonist, via the measurement of light emission. For the measurement, cells (frozen, ready to use) were thawed and resuspended in 10 mL of assay buffer containing 5 μM of coelenterazine h. The cell suspension was then placed in a 10 mL Falcon tube, fixed on a rotating wheel, and incubated overnight at RT° in the dark. Cells were diluted with assay buffer at 5000 cells/20 µL. Agonistic ligands 2 × (50 μL/well), diluted in assay buffer, were prepared in 1/2 area white polystyrene plates, and 50 μL of cell suspension was dispersed on the ligands using the injector. The light emission was recorded for 20 s. Cells were incubated with antagonist for 15 min at RT°. Then, 50 µL of agonist (final concentration equal to 3 × EC80) was injected into the mixture of cells and antagonist and light emission was recorded for 20 s.

The LANCE Ultra cAMP assay as a homogeneous time-resolved fluorescence resonance energy transfer (TR-FRET) immunoassay is designed to measure cAMP produced upon the modulation of adenylyl cyclase activity by GPCRs. The assay is based on the competition between the europium (Eu) chelate-labeled cAMP tracer and sample cAMP for binding sites on cAMP-specific monoclonal antibodies labeled with the ULight dye.

For the measurement, cells (frozen, ready to use) were thawed and resuspended in 4 mL of HBSS 1×. The cell suspension was then placed in a 15 mL Falcon tube and centrifuged for 10 min at 275× *g*. The pellet was resuspended in 1.5 mL of HBSS 1× to determine the cell concentration, and after the next centrifugation the cells were resuspended in stimulation buffer at the appropriate concentration. An antagonist dose–response experiment was performed in 96-well 1/2-area plates using 3000 cells/well, 5 µM forskolin, and 2 nM of reference agonist and antagonist. Cell stimulation was performed for 30 min at RT°, and agonist and antagonists were added simultaneously.

### 4.3. Animals

Experiments were performed on female Wistar rats with an initial body weight of 140–160 g. The animals were housed in plastic cages (2 rats per cage) at a constant room temperature of 22 ± 2 °C, with a 12:12 h light/dark cycle. Water and food were available ad libitum. The randomly established experimental groups consisted of 6 rats for pharmacological studies and 3 rats for pharmacokinetic studies. All experiments were conducted in accordance with the Guide to the Care and Use of Experimental Animals, and were approved by the Jagiellonian University in Krakow Local Ethics Committee for Experiments on Animals (Permission No: 185/2017, 220/2019 and 223A/2019).

### 4.4. In Vivo Studies

#### 4.4.1. Effect of KSK-60, KSK-74, and Phentermine on Changes in Body Weight and Calorie Intake in Non-Obese Rats Fed a Palatable Diet (Model of Excessive Eating)

Female rats were randomly divided into several groups (*n* = 6) and studied for 4 weeks. The groups were as follows: standard diet + vehicle, palatable diet + vehicle, palatable diet + KSK-60 (1 mg/kg), palatable diet + KSK-60 (10 mg/kg), palatable diet + KSK-74 (1 mg/kg), palatable diet + KSK-74 (10 mg/kg), palatable diet + phentermine (7 mg/kg).

The studies for the 1 mg/kg b.w. and 10 mg/kg b.w. doses were performed at different times, so the parameters studied were compared to the respective control groups.

Rats fed a palatable diet had access to a diet consisting of milk chocolate with nuts, cheese, salted peanuts, and 7% condensed milk, and simultaneously to a standard feed (Labofeed B, Morawski Manufacturer Feed, Poland) for 4 weeks [11,21,22]. The water was available ad libitum. Animals had access to palatable products, but also to the standard feed. Most importantly, feeding was not in any way forced, as in other models where animals were fed only a high-fat diet, or temporarily deprived of food (binge eating models). In the case of our model, rats decided themselves when, what, and how much to eat. The weight was evaluated daily. Calorie intake was evaluated three times a week. A palatable diet contained 100 g of peanuts—612 kcal; 100 mL of condensed milk—131 kcal; 100 g of milk chocolate—529 kcal; and 100 g of cheese—325 kcal. The standard diet contained 100 g of feed—280 kcal. Standard diet (fats 8%, carbohydrates 67%, proteins 25%) contained 100 g feed—280 kcal.

The palatable control group (palatable diet + vehicle) received vehicle (1% Tween 80 i.p., daily, for 4 weeks), while palatable test groups were injected i.p. with KSK-60 (1 mg or 10 mg/kg i.p, daily, for 4 weeks), KSK-74 (1 mg/kg or 10 mg/kg i.p, daily, for 4 weeks), or phentermine (7 mg/kg i.p., daily, for 4 weeks), respectively.

The dose of phentermine was selected based on the previous research [23].

On the 31st day, 20 min after i.p. administration of heparin (5000 units/rat) and thiopental (70 mg/kg b.w.), animals were sacrificed, and plasma as well as peritoneal fat pads were collected (food was stopped for 6 h prior to organ, tissue, and plasma collection) for further study.

Figure 14 shows the experiment scheme.

#### 4.4.2. Glucose Tolerance Test

The test was conducted on the 29th day of the experiment. After twenty-eight administrations of the test compounds, the food was stopped for 20 h and then glucose tolerance was tested. Glucose (1 g/kg b.w.) was administered i.p. [19,20]. Blood samples were obtained from the tail vein at the time points 0 (before glucose administration), 30, 60, and 120 min after administration. Glucose levels were measured with a glucometer (ContourTS, Bayer, Leverkusen, Germany, test stripes: ContourTS, Ascensia Diabetes care Poland Sp. z o. o., Poland, REF:84239666). The area under the curve (AUC) was calculated using the trapezoidal rule.

#### 4.4.3. Effect of KSK-60, KSK-74, and Phentermine on Corticosterone and Triglyceride Levels in Rat’s Plasma

Blood was collected from the left carotid artery and then centrifuged at 600× *g* (15 min, 4 °C) to obtain plasma.

To determine plasma triglyceride levels, standard enzymatic and spectrophotometric tests (Cat. No 1-053-A150, Biomaxima S.A., Lublin, Poland) were used. The ELISA Kit (Item No. 501320, Cayman Chemical, Ann Arbor, MI, USA) was used to determine corticosterone levels in plasma.

#### 4.4.4. Effect of KSK-60, KSK-74, and Phentermine on Leptin, Resistin, IL-6, and MCP-1 Concentrations in Adipose Tissue

The frozen peritoneal adipose tissue was weighed, and homogenates were prepared by homogenizing 1 g of tissue in 4 mL of 0.1 M phosphate buffer, pH 7.4, using the IKA-ULTRA-TURRAX T8 homogenizer. The adipose tissue homogenates were then used for biochemical assays.

To determine the levels of leptin, resistin, IL-6, and MCP-1 in adipose tissue, the standard ELISA Kit (Cat. No E0561Ra, Cat. No E0211Ra, Cat. No E0135Ra, Cat. No E0193Ra, Bioassay Technology Laboratory, Shanghai, China) was used. To determine the levels of protein, a standard spectrophotometric test (Cat. No 1-008-B210, Biomaxima S.A. Lublin, Poland) was used.

Biochemical Assays Standard Curves are shown in Appendix A.

#### 4.4.5. Effect of KSK-60, KSK-74, and Phentermine on Spontaneous Activity in Non-Obese Rats Fed a Palatable Diet

The spontaneous activity of the rats was measured on the 1st and 27th days of the treatment with a special RFID system—TraffiCage (TSE-Systems, Berlin, Germany) [21,22]. The animals were subcutaneously implanted with transmitter identification (RFID), which allowed the presence and time spent in different areas of the cage to be counted, and then the data were grouped in a special computer program.

### 4.5. Histological Examinations

Adipose tissue samples were prepared for histology using Masson’s trichrome stain (Merck) by fixation in 4% formaldehyde, dehydration in graded alcohol, and embedded in paraffin. The tissue samples were then sectioned using a microtome of 5 μm thickness. Six random fields from each section were analyzed in a blinded manner with an Olympus BX41 light microscope (Olympus, Tokyo, Japan) at magnifications of ×20, ×40 and ×100, and digital images were captured with an Olympus UC90 color camera. The quantitation of the number of adipocytes was performed in eight fields/slide with ×100 objective using an image analysis system: CellSensDimension (Olympus, Tokyo, Japan). The numerical density of the adipocytes was obtained by counting the total number of adipocytes per high-power field (31,324 mm^2^), and then this number was converted to 0.1 mm^2^. According to stereometric rules, adipocytes tangent to the left and top edges of the image field were not counted. A semi-quantitative scale was used to assess the inflammatory response in adipose tissue. The following were evaluated on a subjective scale: inflammatory cell infiltration between adipocytes (0—normal, 1—few cells, 2—local clusters, and 3—extensive infiltrates), perivascular infiltration of immune cells (0—normal, 1—few cells, and 2—ring of inflammatory cells), widening of the septum (0—normal and 1—widening), and capillary congestion (0—normal and 1—congestion). A sum calculated from independently assessed values was calculated for each animal as an index of inflammation.

### 4.6. Pharmacokinetic Studies

Six rats divided into two experimental groups were used in pharmacokinetic experiments. Three days before the experiment, the rats’ jugular vein was cannulated allowing for multiple blood sampling from a single animal. The investigated compounds were administered i.p. at a single dose of 10 mg/kg. Blood samples (approximately 300 µL) were collected in Eppendorf tubes containing heparin at 5, 15, 30, 60, 120, 240, and 360 min after dosing. Plasma was harvested by centrifuging at 5000× *g* for 10 min and stored at −30 °C until bioanalysis.

Plasma concentrations of KSK-60 or KSK-74 were measured using the liquid chromatography tandem mass spectrometry (LC-MS/MS) method. The samples (50 μL) were deproteinized in a 1:3 (*v*/*v*) ratio with acetonitrile containing an internal standard (IS—pentoxifylline), briefly vortexed, and then centrifuged for 10 min at a speed of 8000× *g* (Eppendorf miniSpin centrifuge). The supernatant was transferred to autosampler vials and a sample volume of 10 μL was injected into the LC-MS/MS system.

Chromatographic separation was performed on the analytical column XBridge™ C18 (3 × 50 mm, 5 µm, Waters, Dublin, Ireland) using the Agilent 1100 HPLC system (Agilent Technologies, Waldbronn, Germany). The mobile phase containing 0.1% formic acid in acetonitrile and 0.1% formic acid in water was run at 0.3 mL/min in gradient mode. Mass spectrometric detection was performed on an Applied Biosystems MDS Sciex (Concord, ON, Canada) API 2000 triple quadrupole mass spectrometer. Electrospray ionization (ESI) in the positive ion mode was used for ion production. The tandem mass spectrometer was operated at unit resolution in the selected reaction monitoring mode (SRM), monitoring the transitions of the protonated molecular ions *m*/*z* 396 to 107 (CE 80 V) for KSK-60, *m*/*z* 424 to 107 (CE 80 V) for KSK-74, and *m*/*z* 279 to 181 (CE 55 V) for IS. Data acquisition and processing were performed using the Applied Biosystems Analyst version 1.6 software. The calibration curves were constructed by plotting the ratio of the peak area of the studied compound to IS versus drug concentration, and generated by weighted (1/x · x) linear regression analysis. The validated quantitation ranges were from 1 to 2000 ng/mL. The calculated accuracy and precision were within the ranges proposed by the guidelines for the validation of bioanalytical methods (FDA, EMA). No significant matrix effect was observed and there were no stability-related problems during the routine analysis of the samples.

Pharmacokinetic parameters were calculated by employing a non-compartmental approach using Monolix version 2019R1 (Antony, France: Lixoft SAS, 2019) software. The area under the mean plasma concentration versus time curve extrapolated to infinity (AUC_0-inf_) was estimated using the log/linear trapezoidal rule. AUMC_0-inf_ was estimated by calculating the total area under the first-moment curve by combining the trapezoid calculation of AUMC0-t and the extrapolated area. The mean residence time (MRT) was calculated from AUMC_0-inf_/AUC_0-inf_. The terminal rate constant (λz) was calculated by log-linear regression of the drug concentration data in the terminal phase, and the terminal half-life (t_1/2_) was calculated as 0.693/λz. Clearance (CL/F) was estimated from the administered dose divided by AUC_0-inf_. The apparent volume of distribution during the terminal phase (Vz/F) was calculated from (CL/F)/λz.

### 4.7. Statistical Analysis

Statistical calculations were performed using the GraphPad Prism 6 program (GraphPad Software, San Diego, CA, USA). The results are presented as arithmetic means with a standard deviation (means ± SD). The normality of the data sets was determined using the Shapiro–Wilk test. Statistical significance was calculated using one-way ANOVA, Tukey’s post hoc or two-way ANOVA, Tukey’s post hoc (body weight) or two-way ANOVA, Bonferroni post hoc (products intake), or Multiple *t*-test (spontaneous activity). Data on the numerical density of adipocytes and the inflammatory index are presented as means ± SD. Comparisons between groups were made using the Kruskal–Wallis test by ranks followed by Dunn post hoc test. Differences were considered statistically significant at * *p* ≤ 0.05, ** *p* ≤ 0.01, *** *p* ≤ 0.001.

## 5. Conclusions

In summary, the compounds we studied were potent H_3_ histamine and sigma-2 receptor ligands with proven efficacy in preventing weight gain in a rat model of excessive eating. The compound KSK-74 significantly compensates for metabolic disturbances that accompany obesity (plasma triglyceride, resistin, and leptin levels), improves glucose tolerance, and protects against adipocyte hypertrophy. Furthermore, KSK-74 inhibits the development of inflammation in obesity-exposed adipose tissue. The in vivo pharmacological activity of the tested ligands appears to correlate with the affinity for the sigma-2 receptors; however, the explanation of this phenomenon requires further and extended research.

## Figures and Tables

**Figure 1 ijms-23-07011-f001:**
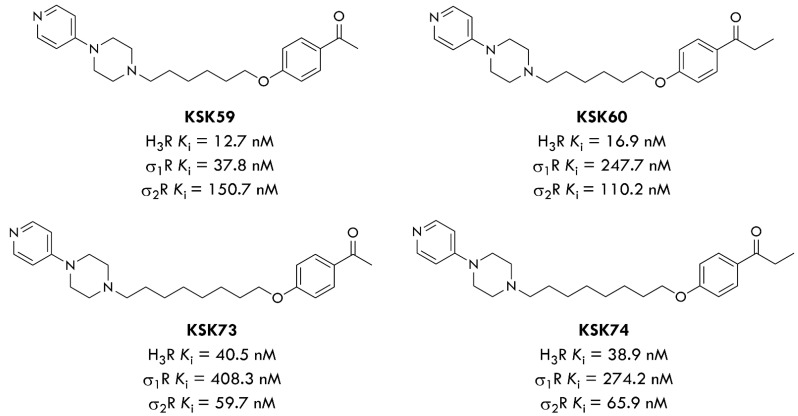
Structures of compounds evaluated in our previous work (KSK-59, KSK-73) and in our present work (KSK-60, KSK-74). Affinity values of the ligands tested at the H_3_R [12] and the sigma-1 and sigma-2 receptors [24]. Structure and purity confirmation data of compounds KSK-60 and KSK-74 were shown in Appendix A.

**Figure 2 ijms-23-07011-f002:**
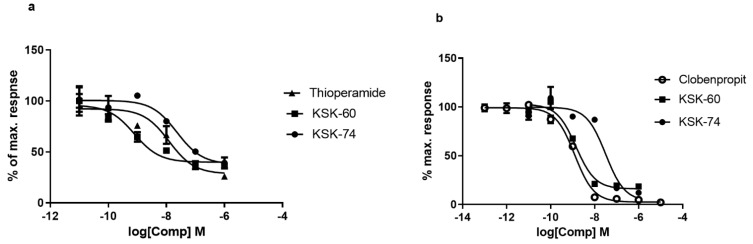
The intrinsic activity at the histamine H_3_ receptor of the tested compounds presented as concentration-dependence curves. The values (%) obtained by two methods, LANCE cAMP (**a**) and Aequoscreen (**b**), are expressed as a percentage of the action of the full agonist (R)-alpha-methylhistamine at the dose of EC_80_ (100%).

**Figure 3 ijms-23-07011-f003:**
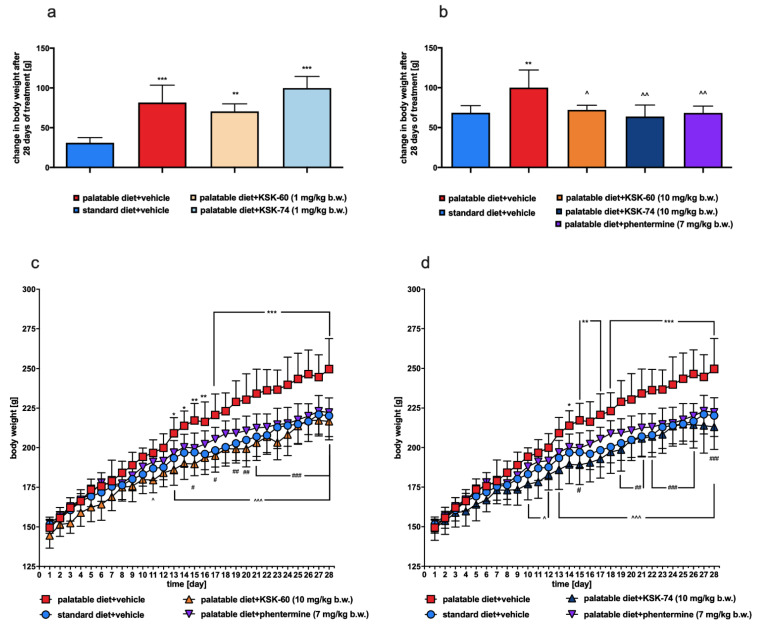
Cumulative changes in body weight (**a**,**b**), body weight during administration of the tested compounds or phentermine, (**c**) KSK-60 (10 mg/kg) and phentermine (7 mg/kg), (**d**) KSK-74 (10 mg/kg) and phentermine (7 mg/kg). Results are expressed as means ± SD, *n* = 6. Multiple comparisons were made using two-way ANOVA, Tukey’s post hoc tests. * Significant against standard diet + vehicle group vs. palatable diet + vehicle group; ^ significant against the tested compound administered group vs. palatable diet + vehicle group; # significant against palatable diet + phentermine group vs. palatable diet + vehicle group; *, ^, # *p* < 0.05, **, ^^, ## *p* < 0.01, ***, ^^^, ### *p* < 0.001.

**Figure 4 ijms-23-07011-f004:**
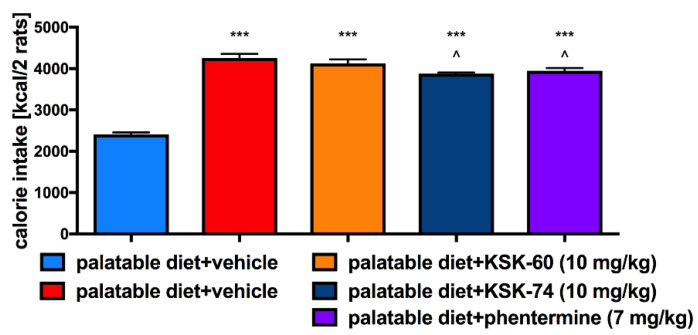
Effect of administration of the tested compounds or phentermine on calorie intake compared to control groups. Results are expressed as means ± SD, *n* = 3. Comparisons were made using the Kruskal–Wallis test followed by the Dunn post hoc test * Significant against control rats fed standard diet; * *p* < 0.05, ** *p* < 0.01.

**Figure 5 ijms-23-07011-f005:**
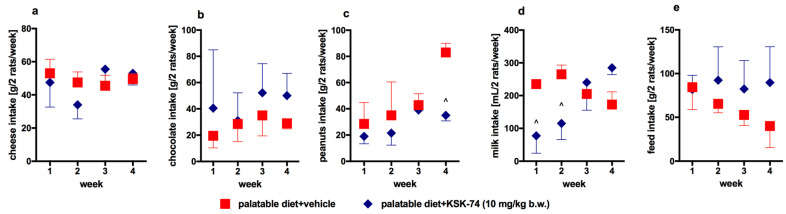
Amount of particular taste products’ intake: (**a**) cheese, (**b**) chocolate, (**c**) peanuts, (**d**) milk, (**e**) feed, by rats from group treated with KSK-74 compared to rats from control group fed palatable feed. Results are expressed as means ± SD, *n* = 3. Multiple comparisons were made using two-way ANOVA, Bonferroni’s post hoc tests. ^ Significant against the tested compound administered group vs. palatable diet + vehicle group; ^ *p* < 0.05.

**Figure 6 ijms-23-07011-f006:**
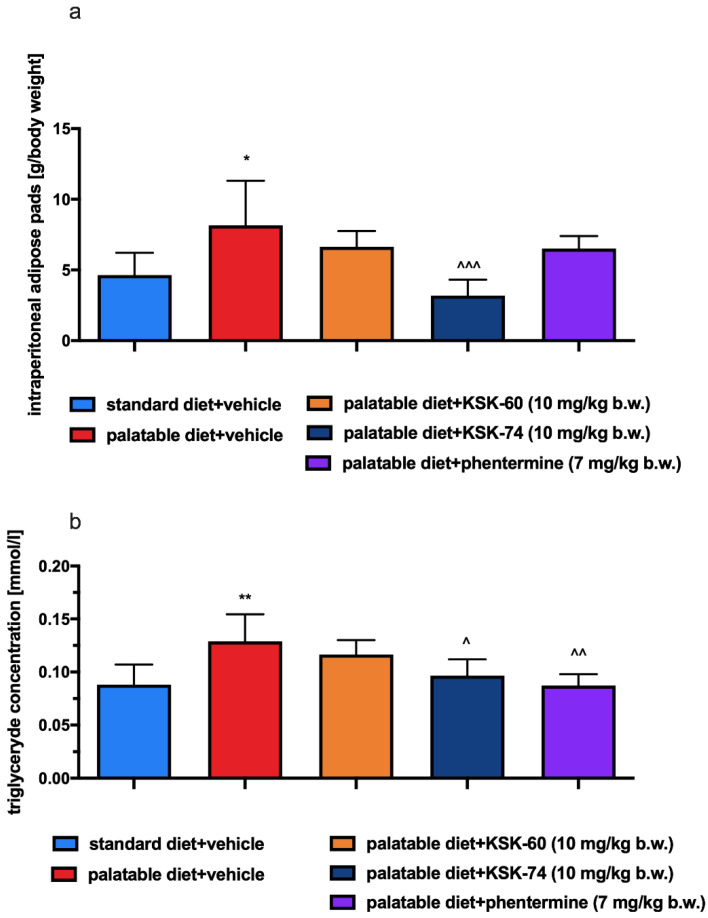
Mass of adipose pads at the end of the experiment (**a**), effect of the tested compounds or phentermine on the plasma level of triglyceride (**b**). Results are expressed as means ± SD, *n* = 6. Comparisons were made using one-way ANOVA, Tukey’s post hoc tests. * vs. standard diet + vehicle group; ^ vs. palatable diet + vehicle group; ^, * *p* < 0.05, ^^, ** *p* < 0.01, ^^^ *p* < 0.001.

**Figure 7 ijms-23-07011-f007:**
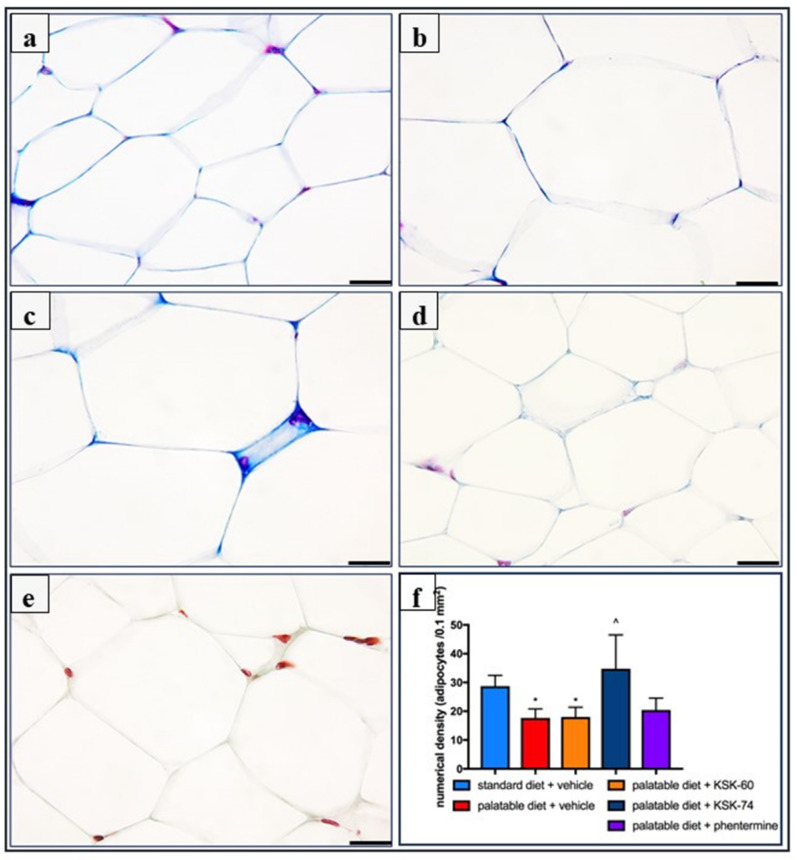
Adipocytes numerical density in Masson’s trichrome staining of adipose tissue from the studied groups of rats. Representative microphotographs of tissue sections from the standard diet + vehicle group showing the normal architecture of adipocytes (**a**); palatable diet + vehicle (**b**) and palatable diet + KSK-60 10 mg/kg (**c**) showing a decrease in the number of adipocytes per 0.1 mm^2^; palatable diet + KSK-74 10 mg/kg (**d**), which has more adipocytes compared to palatable diet + vehicle; and palatable diet + phentermine 7 mg/kg (**e**). Bar = 20 mm. (**f**) Adipocytes’ numerical density expressed as means ± SD, *n* = 6. Comparisons were made using the Kruskal–Wallis test followed by the Dunn post hoc test; * significant vs. standard diet + vehicle group; ^ significant vs. palatable diet + vehicle group * ^ *p* < 0.05.

**Figure 8 ijms-23-07011-f008:**
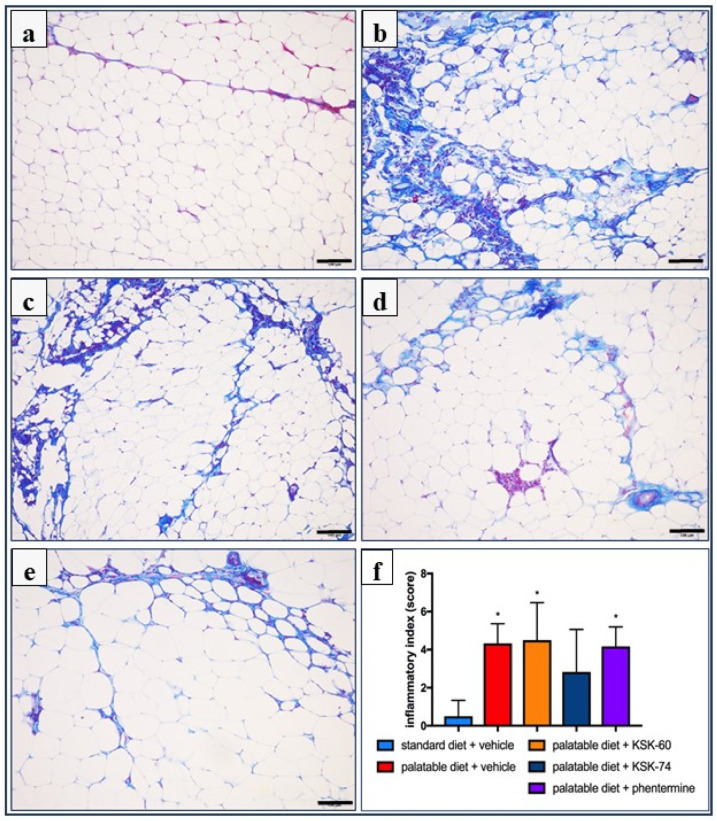
Representative histological pictures of Masson’s trichrome-stained adipose tissue sections (bar = 100 mm). (**a**) Standard diet + vehicle group, (**b**) palatable diet + vehicle, (**c**) palatable diet + KSK-60 10 mg/kg, (**d**) palatable diet + KSK-74 10 mg/kg, (**e**) palatable diet + phentermine 7 mg/kg. Many inflammatory cells infiltrating the area of the adipocytes and trichrome-positive fibrotic streaks are visible in the adipose tissue of all studied groups, except the rats in the standard diet + vehicle group. (**f**) Inflammatory index expressed as means ± SD, *n* = 6. Comparisons were made using the Kruskal–Wallis test followed by the Dunn post hoc test; * significant vs. standard diet + vehicle group; * *p* < 0.05.

**Figure 9 ijms-23-07011-f009:**
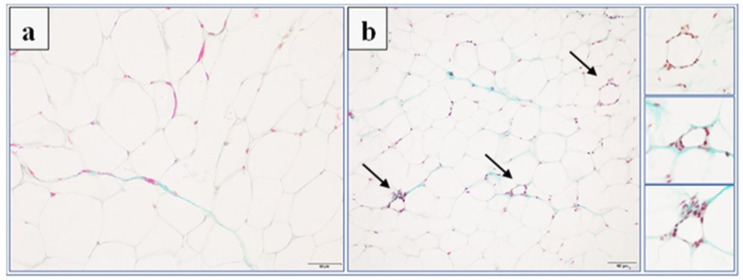
Histological section of adipose tissue from the palatable diet + phentermine 7 mg/kg group stained with Masson’s trichrome showing capillary congestion (**a**) and the frequent appearance of crown-like structures (arrow) formed by mononuclear cells surrounding a presumably dead adipocyte (**b**). Bar = 50 mm (**a**) and 100 mm (**b**).

**Figure 10 ijms-23-07011-f010:**
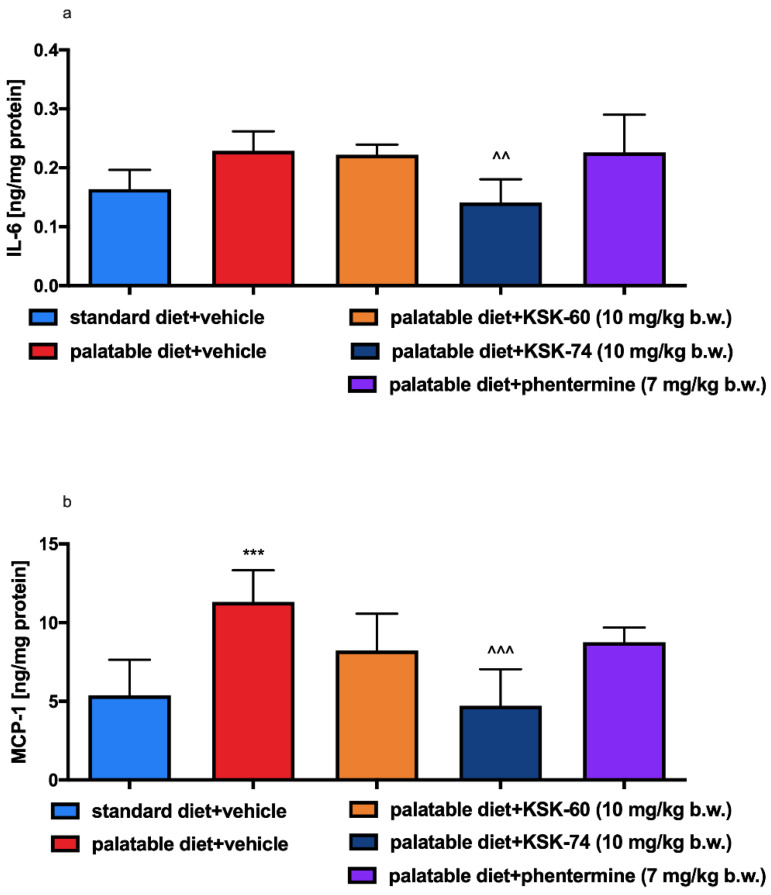
Effect of administration of the tested compounds or phentermine on adipose tissue levels of (**a**) IL-6 and (**b**) MCP-1. Results are expressed as means ± SD, *n* = 6. Comparisons were made using one-way ANOVA, Tukey’s post hoc test; * significant vs. standard diet + vehicle group; ^ significant vs. palatable diet + vehicle group ^^ *p* < 0.01, ***, ^^^ *p* < 0.001.

**Figure 11 ijms-23-07011-f011:**
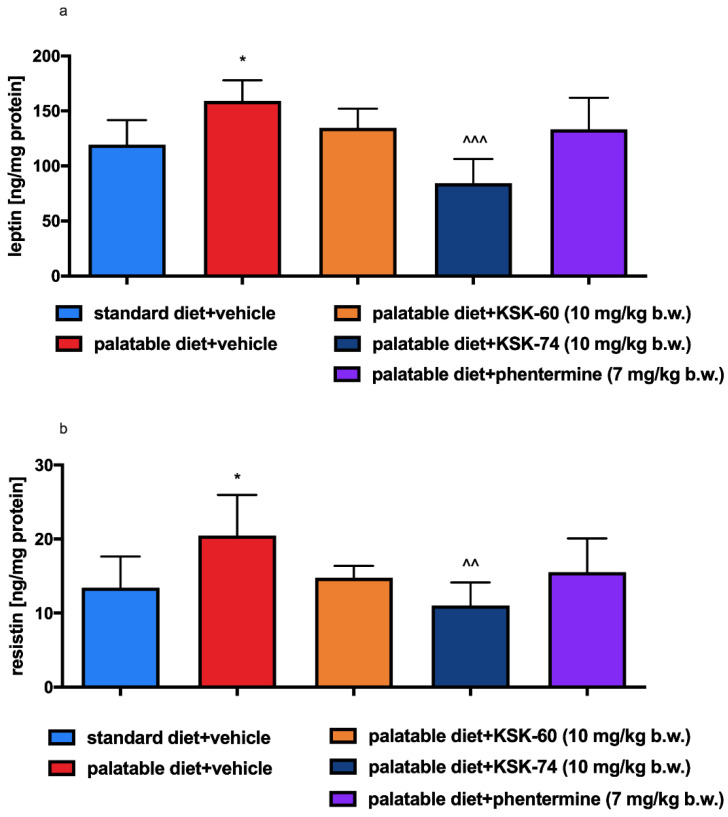
Effect of the tested compounds or phentermine on adipose tissue levels of (**a**) leptin and (**b**) resistin. Results are expressed as means ± SD, *n* = 6. Comparisons were performed by one-way ANOVA, Tukey’s post hoc test; * significant vs. standard diet + vehicle group; ^ significant vs. palatable diet + vehicle group * *p* < 0.05, ^^ *p* < 0.01, ^^^ *p* < 0.001.

**Figure 12 ijms-23-07011-f012:**
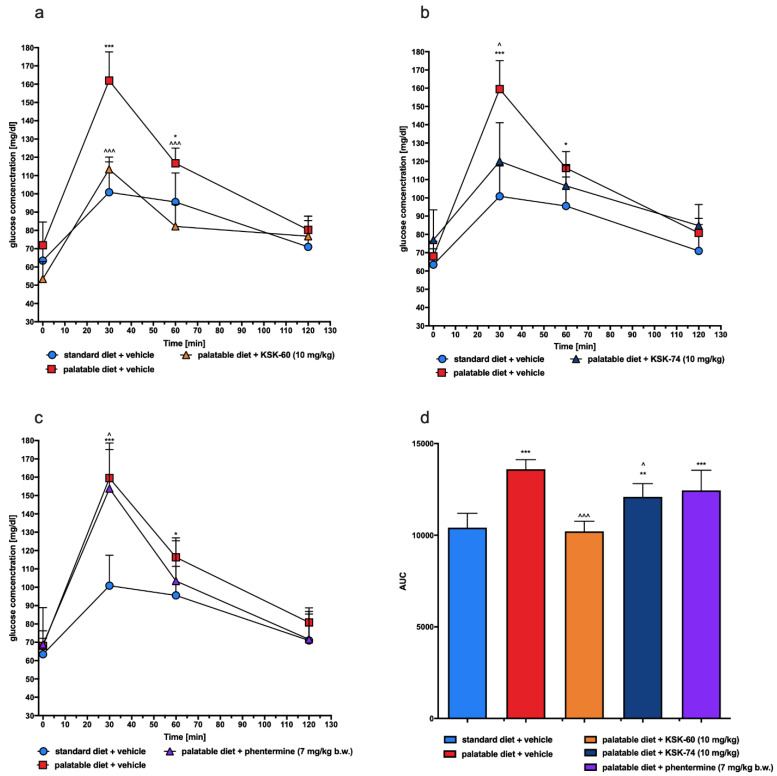
Glucose tolerance test. Results are expressed as means ± SD, *n* = 6. (**a**–**c**) Intraperitoneal glucose tolerance test (IPGTT): multiple comparisons were made using two-way ANOVA, Tukey’s post hoc tests. (**d**) Area under the IPGTT curve: comparisons were made using one-way ANOVA, Tukey’s post hoc test. * Significant vs. standard diet + vehicle group; ^ significant vs. palatable diet + vehicle group; *, ^ < 0.05, ** *p* < 0.01, ***, ^^^ *p* < 0.001.

**Figure 13 ijms-23-07011-f013:**
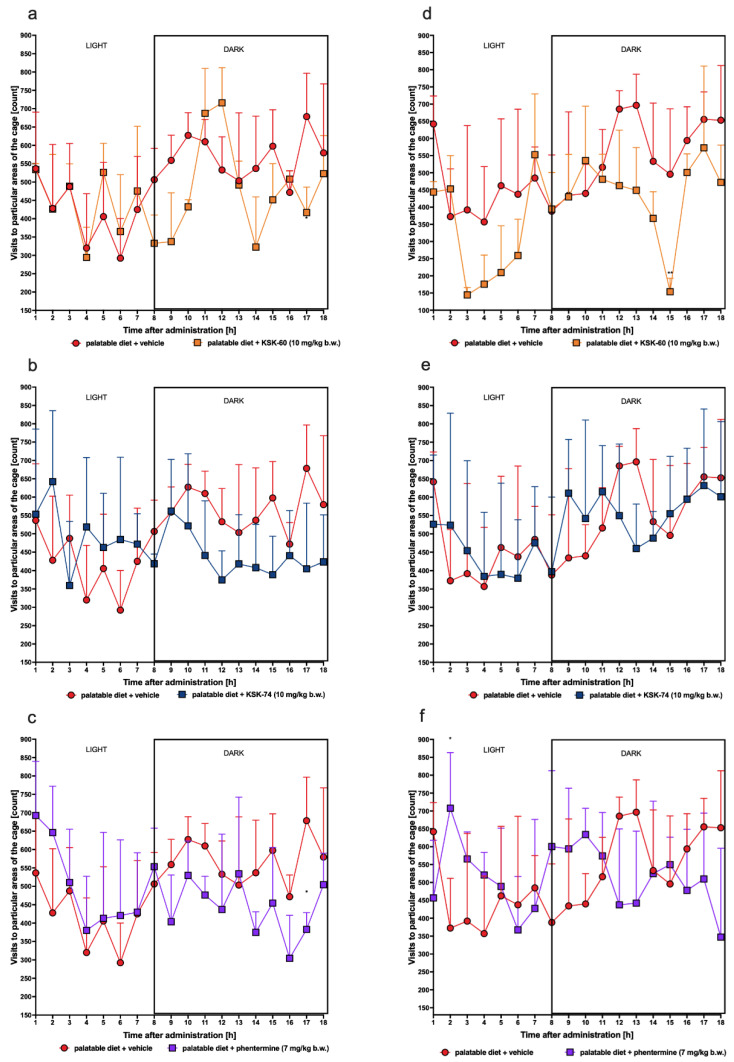
Spontaneous activity after the 1st and 27th administrations of the tested compounds. Results are expressed as means ± SD, *n* = 6. Comparisons were made using Multiple *t*-test; (**a**) palatable diet + vehicle vs. palatable diet + KSK-60 after the 1st administration; (**b**) palatable diet + vehicle vs. palatable diet + KSK-74 after the 1st administration; (**c**) palatable diet + vehicle vs. palatable diet + phentermine after the 1st administration; (**d**) palatable diet + vehicle vs. palatable diet + KSK-60 after the 27th administration; (**e**) palatable diet + vehicle vs. palatable diet + KSK-74 after the 27th administration; (**f**) palatable diet + vehicle vs. palatable diet + phentermine after the 27th administration; * *p* < 0.05, ** *p* < 0.01.

**Figure 14 ijms-23-07011-f014:**
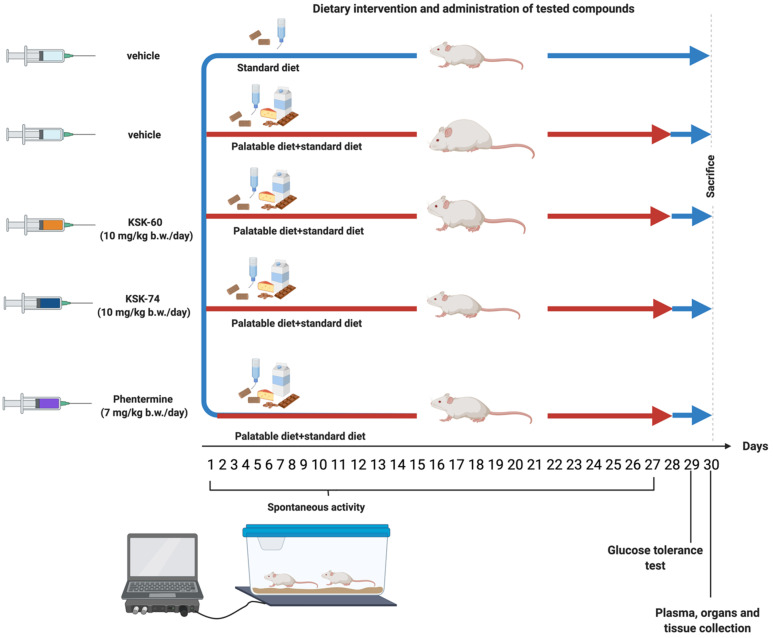
Experiment scheme.

**Table 1 ijms-23-07011-t001:** The IC_50_ values for the antagonist dose–response with the reference agonist ((R)-alpha-methylhistamine) at a final concentration equivalent to EC_80_ obtained by two methods: LANCE Ultra cAMP and Aequoscreen.

Compound	LANCE Ultra cAMP IC_50_ [nM]	Aequoscreen IC_50_ [nM]
(R)-alpha-methylhistamine	1.05 ± 0.1	10.6 ± 2.2
Clobenpropit	-	1.1 ± 0.3
Thioperamide	14.52 ± 3.2	-
KSK-60	0.78 ± 0.1	1.49 ± 0.2
KSK-74	23.22 ± 3.6	31.9 ± 0.5

**Table 2 ijms-23-07011-t002:** Inflammation in adipose tissues. The number of rats in each group, with the appropriate histological grade for the independently assessed components of the inflammatory.

	Inflammatory Cell Infiltration				Perivascular Infiltration			Widening of the Septum		Capillary Congestion	
Grade:	0-Normal	1-Few Cells	2-Local Clusters	3-Extensive Infiltrates	0-Normal	1-Few Cells	2-Ring of Cells	0-Normal	1-Widening	0-Normal	1-Congestion
Standard diet + vehicle	4	2	-	-	6	-	-	5	1	6	-
Palatable diet + vehicle	-	-	5	1	2	2	2	-	6	5	1
Palatable diet + KSK60	-	2	4	-	-	2	4	1	5	2	4
Palatable diet + KSK74	1	3	2	-	2	1	3	3	3	5	1
Palatable diet + phentermine	-	4	2	-	1	4	1	1	5	-	6

**Table 3 ijms-23-07011-t003:** Estimated pharmacokinetic parameters (non-compartmental analysis) of the investigated compounds calculated from mean rat plasma concentration values (*n* = 3) after administration of a single i.p. dose of 10 mg/kg.

Parameter	KSK-60	KSK-74
C_max_ [µg/L]	109	105
t_max_ [h]	0.083	0.083
λ_z_ [h^−1^]	0.26	0.25
t_0.5λz_ [h]	2.7	2.71
CL_S_/F [l/h/kg]	69.98	131.06
AUC_0-inf_ [mg∙h/L]	514.40	274.67
V_z_/F [l/kg]	272.63	513.14
MRT [h]	3.28	2.45

C_max_—maximal concentration, t_max_—time to reach maximal concentration, λz—terminal elimination rate constant, t_0.5λz_—half-life, CLs—clearance, AUC_0-inf_—area under the concentration–time curve extrapolated to infinity, Vz—volume of distribution, MRT—mean residence time, F—bioavailability.

## Data Availability

The data presented in this study are available on request from the corresponding author.

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
