# Peer review of "KSK-74: Dual Histamine H3 and Sigma-2 Receptor Ligand with Anti-Obesity Potential"

_ijms, 2022, doi:10.3390/ijms23137011_

Round 1

Reviewer 1 Report

Mika et al., characterize the anti-obesity effects of two new molecules, KSK-60 and KSK-74. They show that administration of each compound to rats that were fed with obesogenic diet reduces weight gain and accordingly improves glucose tolerance, plasma triglyceride, and reduces adipocytes hypertrophy. No major effects were observed on adipose tissue inflammation. The authors also nicely characterize the pharmacokinetic properties of KSK-60 and KSK-74.

Major points:

1.       It will be important to add more information on the diet used. Specifically, how many calories are derived from fat and carbohydrates

2.       Is there any toxicity associated with the injection of the compounds? What is the ALT/AST ratio following injection?  

3.       Most importantly, is there any effect on eating behavior?

Reviewer 2 Report

The investigative group of the manuscript “KSK-74 – dual histamine H3 and sigma-2 receptor ligand with anti-obesity potential” evaluated the pharmacological properties of the KSK-60 and KSK-74 on an obese female rat model. Although, it looks like the authors have exaggerated the findings to derive a meaningful title. There is no underlying mechanistic study to support the said molecules have H3 and sigma-2 receptor interaction in-vitro and in-vivo experiments. The authors have merely looked at the biochemical and histopathological perturbation without investigating the gene or protein level effect. The authors need to address the following concerns in the revised version of the manuscript.

1.     Please cite the most recent literature on the epidemiology of obesity in the introduction section.

1.     Authors in the results section “2.5. Effect of KSK-60, KSK-74, or Phentermine on the presence of pathological features of adipose tissue inflammation” mentioned that no leukocyte infiltration was observed in the adipose tissue (Figure 7). I wonder how they check for leukocyte infiltration with Masson’s trichrome stain. Ideally, Hematoxylin and Eosin are recommended for the same.

2.     How did you analyze the inflammatory index? Share the detailed protocol for the same.

3.     Did you also study the biological effect of any other dose except 1mg/kg, 10 mg/kg of KSK-60, and KSK-74?

4.     Please include the NMR and LC-MS data to identify and purity the compounds in the supplementary.

5.     The authors should have studied KSK-60 and KSK-74 in male rats since female rats have an intuitive protective mechanism.

6.     Share the catalog no. of all the commercial kits used in the study and the standard curves.

7.     Mention the complete details description of the statistical software used.

8.     Include a separate heading to share the limitations of the current study.

Round 2

Reviewer 1 Report

I have no further requests

Reviewer 2 Report

The authors of the revised manuscript “KSK-74 – dual histamine H3 and sigma-2 receptor ligand with anti-obesity potential” justified all my concerns legitimately and have incorporated my suggestions in the revised document. I hope that the editors will now consider it for publication.